# Controlling Gut Microbiota by Twendee X^®^ May Contribute to Dementia Prevention

**DOI:** 10.3390/ijms242316642

**Published:** 2023-11-23

**Authors:** Fukka You, Yoshiaki Harakawa, Toshikazu Yoshikawa, Haruhiko Inufusa

**Affiliations:** 1Division of Anti-Oxidant Research, Life Science Research Center, Gifu University, Yanagito 1-1, Gifu 501-1194, Japan; y@antioxidantres.jp (F.Y.); harakawa@antioxidantres.jp (Y.H.); 2Anti-Oxidant Research Laboratory, Louis Pasteur Center for Medical Research, Tanakamonzen-cho 103-5, Kyoto 606-8225, Japan; 3Louis Pasteur Center for Medical Research, Tanakamonzen-cho 103-5, Kyoto 606-8225, Japan; toshi@yoshikawalab.jp; 4Department of Cellular Regenerative Medicine, Graduate School of Medical Science, Kyoto Prefectural University of Medicine, Kajii-cho, Kawaramachi-Hirokoji, Kyoto 602-8566, Japan

**Keywords:** Alzheimer’s disease, dementia, Twendee X, microbiome, gut microbiota, oxidative stress, quality of life, antioxidant

## Abstract

The human gut microbiota (GM) is a complex and dynamic ecosystem that hosts trillions of commensal and potentially pathogenic microorganisms. It is crucial in protecting humans from pathogens and in maintaining immune and metabolic homeostasis. Numerous studies have demonstrated that GM has a significant impact on health and disease, including Alzheimer’s disease (AD). AD is a progressive neurodegenerative disorder characterized by impaired short-term memory and cognitive deficits. Patients with AD have been reported to exhibit abnormalities in GM density and species composition. Oxidative stress (OS) has been implicated in the onset and progression of AD; however, the relationship between OS and gut microbiota in AD onset and progression is not clear. Twendee X^®^ (TwX), an oral supplement consisting of eight active ingredients, has been shown to prevent dementia in mild cognitive impairment (MCI) in humans and substantially improve cognitive impairment in mouse models of AD. This positive effect is achieved through the potency of the combined antioxidants that regulate OS; therefore, similar results cannot be achieved by a single antioxidant ingredient. To examine the impact of long-term OS elevation, as seen in AD on the body and GM, we examined GM alterations during the initial OS elevation using a two-week OS loading rat model, and examined the effects of TwX on OS and GM. Furthermore, using a questionnaire survey and fecal samples, we analyzed the impact of TwX on healthy individuals’ gut bacteria and the associated effect on their quality of life (QOL). TwX was found to increase the number of bacteria species and their diversity in GM, as well as butyrate-producing bacteria, which tend to be reduced in AD patients. Additionally, TwX improved defecation condition and QOL. The gut bacteria function as part of the homeostatic function during OS elevation, and the prophylactic administration of TwX strengthened this function. The results suggest that the preventative effect of TwX on dementia may involve the GM, in addition to the other previously demonstrated effects.

## 1. Introduction

The human intestinal tract is home to thousands of species and over 100 trillion bacteria, which maintain a symbiotic relationship. The host provides the gut bacteria with a habitat and nutrients, and, in turn, gut bacteria participate in the metabolism of bile acids and nutrients such as amino acids and carbohydrates, the production of metabolites such as short-chain fatty acids, vitamin synthesis, infection defense against pathogens, and the differentiation and maturation of intestinal epithelial cells and immune cells.

Although the gut microbiota (GM) remains relatively stable during adulthood, individual differences increase with age [1,2], as the composition of the GM undergoes significant alterations [3], including a decline in diversity and stability [4,5,6]. The increased permeability of the intestine and blood–brain barrier induced by GM abnormalities may mediate or influence the etiology of Alzheimer’s disease (AD) and other neurodegenerative diseases, especially those associated with aging [7]. A growing number of reports suggest that increased intestine permeability is implicated in AD, Parkinson’s disease (PD), and other neurodegenerative diseases, as well as acute central nervous system (CNS) damage such as ischemic stroke [8,9]. In addition to this, AD is now recognized as an oxidative stress (OS) disease [10,11,12,13,14], and its pathogenesis is the consequence of extended OS elevation for at least 20 years. To examine the impact of long-term OS elevation, we theorized that investigating changes in GM and daily life during the initial period of elevation would provide key insights into preventing long-term effects.

Twendee X^®^ (TwX), is a supplement that combines eight active ingredients [15,16] and has passed all safety tests (chromosomal aberration, toxicity, and mutation studies) required for pharmaceutical products. The results from a multicenter, randomized, double-blind, and placebo-controlled clinical intervention trial have shown that TwX is effective in preventing dementia in Japanese patients with mild cognitive impairment (MCI) [17]. TwX also improves motor coordination and working memory, while also restoring hippocampal neuron loss in a mouse model of AD with chronic cerebral hypoperfusion (CCH + APP23 mice). Additionally, notable improvements in cognitive deficits, reductions in Aβ pathology and neuronal loss, and an alleviation of neuroinflammation and OS were observed [18,19]. In an ischemic stroke mouse model, pretreatment with TwX (20 mg/kg/day) for 14 days resulted in reduced infarct size and reduced expression of OS markers, tumor necrosis factor-α (TNF-α), and inflammation markers [20]. In mice with vitamin-E-deficiency-induced OS that induced impaired cognitive function and coordination, TwX significantly improved cognitive function and coordination, while also resulting in significant increases in brain-derived neurotrophic factor and nerve growth factor levels [21]. These effects of TwX on neurodegenerative diseases have also been shown to include the protective effects of TwX on mitochondria through its strong antioxidant capacity, which cannot be achieved by a single antioxidant ingredient, and the concomitant increase in ATP production, reduction in blood OS and maintenance of autophagy function, telomere elongation, and maintenance of neurogenesis [22]. Additionally, recent research has indicated brain–gut correlations in dementia. Although the effects of TwX on OS, mitochondria, and cranial nerves have been elucidated, its effects on the brain–gut correlation is currently unknown. Furthermore, the effects of elevated OS on gut microbiota have yet to be elucidated. Based on the results of previous studies, TwX may have preventative effects on dementia by stimulating changes in GM. In this paper, we examined the effects of TwX on GM in a rat model for short-term OS exposure, which mimics the initial stages of long-term elevated OS. Furthermore, this paper discusses changes in quality of life (QOL) and GM among healthy individuals who took TwX, based on the outcomes of a questionnaire survey.

## 2. Results and Discussion

### 2.1. OS and Gut Microbiota

Orthophenyl phenol (OPP) is used as a fungicide and preservative for fruits and fruit products. OPP is known to induce reactive oxygen species (ROS) in the body, such as causing cancer in the urinary tract of rats [23,24,25]. Since there are opportunities to ingest OPP residues in the fruit peel through foods, the OPP-induced OS model was used as a natural model for elevated OS. In a study using a rat OPP OS-induced model, blood OS was measured by d-ROMs test (d-ROMs), blood antioxidant capacity by BAP test (BAP), and body antioxidant capacity by OXY adsorbent test (OXY). After administering OPP for 2 weeks (1 μg/kg/day), both BAP and OXY increased in the rats; however, d-ROMs showed higher levels than in normal rats (normal group). In the group administered with TwX (20 mg/kg/day) alone (TwX group), both BAP and OXY were elevated (with significant differences in BAP) and d-ROMs were significantly reduced compared to normal healthy rats. The TwX + OPP group, administered with TwX one week prior to OPP administration to assess its preventive effect, exhibited marginally reduced BAP compared to the normal, OPP, and TwX groups. However, these rats maintained elevated OXY levels and significantly reduced the rate of increase in OPP-induced OS (Figure 1A–C).

The increased BAP and OXY in the OPP group likely indicates normal homeostatic function in response to heightened OS; however, no accompanying reduction in OS was observed. Although OS is deeply involved in various pathological conditions, elevated OS exceeding 20 years, as observed in AD, may shape the pathogenesis of the disease [26]. Sustained elevated OS, regardless of duration, damages the body’s homeostatic functions, which may eventually lead to long-term elevated OS. In the TwX + OPP group, where TwX was administered prophylactically, OS levels decreased. In the previous study, TwX also reduced ROS in cells and mitochondria while increasing SOD, an antioxidant enzyme, as well as aiding in the scavenging of free radicals [22]. The results of the TwX group showed that increasing the available antioxidants in the blood and body can reduce OS in the blood. These results provide evidence that the prophylactic administration of TwX can maintain lower levels of OS even if ROS is continuously produced in the body.

After administering OPP, fecal samples were collected from the ileal region of the rats for GM analysis. We observed that the diversity of bacterial species in the OPP group tended to be slightly lower than that in the normal group. However, the increase in OS had no major effect. In contrast, the TwX group and the TwX + OPP group both showed higher bacterial diversity than the normal group, with the TwX + OPP group showing the most diversity (Figure 2A,B). The findings suggest that bacterial diversity also plays a critical role in mitigating OS. Diversity is increased by taking only the antioxidant TwX. In other words, some bacteria are incapable of tolerating prolonged exposure to OS. This is consistent with the tendency of GM composition to alter significantly, such as a decrease in diversity with old age, which is the greatest risk factor for AD [4,5,6]. Therefore, increasing bacterial diversity is thought to be a way to prevent dementia.

Some intestinal bacteria produce short-chain fatty acids (SCFA: butyric acid, acetic acid, and propionic acid). Consuming plant foods rich in soluble dietary fiber has been reported to influence intestinal bacteria to produce SCFA [27].

Regarding butyrate-producing bacteria (genera *Roseburia*, *Coprococcus*, *Lachnospira*, *Clostridium*, and *unclassified Lachnosiraceae*) [28,29], the TwX, OPP and TwX + OPP groups showed a higher increase in these bacteria than the normal group (Figure 3A). This trend was particularly pronounced in the TwX + OPP group. For *Akkermansia* bacteria, which also have anti-inflammatory activity, there was no significant difference in the TwX group and the OPP group compared to the normal group, but the TwX + OPP group showed a substantial increase in these bacteria (Figure 3B).

OS is an important process in the pathogenesis of AD.

OS is an imbalance between ROS production and the body’s defense mechanisms, which leads to chronic inflammation. OS activates various transcription factors and causes the expression of several genes involved in inflammatory pathways. Inflammation induced by OS is the cause of many chronic diseases [30].

Butyric acid has anti-inflammatory properties [27], and has been reported to reduce the detrimental effects caused by ROS in a healthy and balanced gut [31]; this includes a positive role for the GM in reducing ROS production via SCFAs such as N-butyric acid [32]. In addition, the genus *Akkermansia* is a beneficial bacteria, especially the outer membrane protein (Amuc_1100) of *A. muciniphila*, which enhances intestinal barrier function via TLR-2 and ameliorates inflammation, thereby inhibiting intestinal barrier dysfunction in obesity and diabetes [30]. The increase in butyrate-producing bacteria in the OPP group may also be due to normal homeostatic function in response to increased OS. Administering TwX alone resulted in higher GM diversity and butyric acid bacteria, but no significant changes in the *Akkermansia* bacteria. However, prophylactic TwX treatment not only kept BAP and OXY levels high, but also increased GM diversity and the abundance of both butyrate-producing bacteria and *Akkermansia* spp., even in the presence of persistent ROS exposure; this ultimately eliminated ROS and improved inflammation. Evidently, the GM plays a role in the initial control of OS in the host. We found that TwX also acted on GM, thereby reducing OS, which is one of the mechanisms through which TwX helps to prevent dementia. However, the full mechanism leading to this result is not yet clear and should be further investigated. The prophylactic administration of TwX in the early stage of elevated OS may prevent AD and other long-term effects on the body.

### 2.2. Changes in the Gut Microbiota in Healthy Individuals

To investigate the effects of TwX on GM in healthy individuals, participants were recruited to complete a questionnaire survey and supply fecal samples (Eyes, Inc., Tokyo, Japan). Using the website, participants could access the purpose of the study, TwX dosing instructions, and stool sampling instructions. Each participant completed a pre-survey to self-assess their lifestyle, defecation status, and physical condition, and completion of the pre-survey was considered as consent.

There were 114 participants consisting of 57 pairs of healthy males and females, aged 35 years or older (Figure 4A). For GM analysis, fecal samples were analyzed by 16S rRNA sequencing (Takara Bio Inc., Shiga, Japan) before and after taking TwX (13.25 mg/kg/day).

The participants’ diet included a relatively high intake of vegetables and dairy products, and most preferred meat to fish. Almost half of the participants consumed highly processed foods (junk) daily, and females were more likely to consume sweets, high-sugar foods and dairy products (Figure 4C). More than half of both males and females tended to drink alcohol, daily or occasionally, with daily alcohol consumption being higher among males. Both males and females exercised infrequently (Figure 4B).

Participants took TwX while maintaining their usual lifestyle.

After taking TwX for 4 weeks, the number and diversity of GM bacteria increased (Figure 5A,B) while the *Firmicutes*/*Bacteroidetes* (F/B) ratio, which is considered an index of obesity, decreased (Figure 5C).

Butyrate-producing bacteria [28] increased while propionate-producing bacteria [28] decreased after participants took TwX (Figure 6B,C). The relatively abundant bacteria in the GM of Japanese with long lifespans [29] also increased after taking TwX (Figure 6A).

With age, one of the greatest risk factors for AD, the quality of the GM environment tends to decline. For example, an increase in facultative anaerobes and a decrease in beneficial bacteria such as anaerobic *lactobacilli* and *bifidobacteria* have been reported. These changes, along with the decreased species diversity in most bacterial groups and changes in digestive physiology, such as diet and intestinal transit time, may lead to increased putrefaction and greater susceptibility to disease in the colon [3].

The two most important bacterial phyla in the GI tract, *Firmicutes* and *Bacteroidetes*, comprise 90% of the GM [33]. The F/B ratio has been proven to impact normal intestinal homeostasis; an increased F/B ratio for example, is often observed in obesity, one of the risk factors for AD [34]. TwX has been observed to increase the number and diversity of bacterial species in GM and decrease the F/B ratio.

SCFAs have been studied as by-products of bacterial fermentation following the ingestion of soluble fiber, with acetate, propionate, and butyrate being the major bacterial products in the colon [35]. SCFAs have been reported to be released into the bloodstream and may reach the brain [36]. Although the molecular structure of propionate and butyrate differ by a single carbon, they have opposite effects in brain cells. Butyrate has an anti-inflammatory effect, while propionate has been reported to act on neuroinflammation [27]. In addition, butyrate has been reported to improve learning disabilities, as well as to improve the dendritic spine density of hippocampal neurons in Tg2576 mice, a mouse model of AD [37]. If left untreated, abnormalities in the GM may lead to microglial activation, BBB disruption, and subsequent systemic inflammation that determines pathogen and immune cell crossover [38].

In this study, TwX resulted in an increase in butyrate-producing bacteria and a decrease in propionate-producing bacteria. TwX also increased the number of beneficial bacteria and decreased the number of harmful bacteria in the intestines of individuals with relatively high levels of harmful bacteria in their GM (https://www.twendee.com/files/twendee/study-results/twendee-gut-microbiome.pdf (accessed on 24 October 2023)). Additionally, TwX may also contribute to longevity, as it increases the relatively abundant butyric acid bacteria found in Japanese individuals with long lifespans. This result can be linked to the telomere elongation effect demonstrated in a previous study [22].

The number of GM species is vast, and the species may have many unknown species and undiscovered functions. Therefore, it is difficult to accurately predict the direct impact that changes in the GM can have on the body. However, based on the results of this survey, we believe that TwX may have a preventive effect on AD by regulating GM balance and inhibiting the intestinal changes that occur before the onset of AD.

### 2.3. Effects on Quality of Life

The defecation status, stool condition, and physical condition before and after taking TwX were calculated from the responses to a self-assessment before and after taking TwX (Eyes, Inc.).

The defecation status of males before taking TwX was that 84% had pleasant bowel movements (72% had a smooth defecation every day, 10% had a smooth defecation once every 2 days, and 2% had relatively smooth bowel movements generally), while the remaining 16% were constipated for three or more days (Figure 7A Before). The stools were normal (sausage shape) in 53%, lumpy in 12%, separate hard lumps in 10%, soft stools in 23%, and liquid stools in 2% (Figure 8A Before). The defecation status of the females before taking TwX showed that 79% had pleasant bowel movements (42% had smooth defecation daily, 33% had a smooth defecation every 2 days, and 4% had relatively smooth bowel movements generally). The remaining 21% were constipated for more than 3 days, of which 2% were only able to defecate with the use of laxatives (Figure 7B Before). Stools were normal (sausage shape) in 58%, lumpy in 7%, separate hard lumps in 19%, and soft stools in 16% (Figure 8B Before). The defecation status prior to taking TwX was relatively better for males in terms of bowel movements, but better for females in terms of stool condition. Since the participants in this survey had no serious health problems, the majority had good stools quality and smooth bowel movements, with no signs of severe constipation. After taking TwX for 4 weeks, 95% of males had a pleasant bowel movement. This was an improvement: 76% had a smooth defecation every day, 14% had a smooth defecation once every 2 days, 5% had relatively smooth bowel movements generally (Figure 7A After), and constipation for more than 3 days decreased to 5% (Figure 7B After). The number of normal stools (sausage shape) increased to 60%, lumpy (7%) and separate hard lump stools (7%) both decreased, soft stools were present in 26%, and there was no liquid stool. (Figure 8A After). The defecation status of females after taking TwX similarly improved, with 88% having a pleasant bowel movement (67% having a smooth defecation daily, 18% having a smooth defecation once every 2 days, and 3% having a relatively smooth bowel movement), and constipation for more than three days decreased to 7%. However, the percentage of those who could defecate only with the use of laxatives rose to 5% (Figure 7B After). For stool condition, normal stools (sausage shape) were improved to 75%, and lumpy stools (7%), separate hard lump (9%), and soft stools (9%) were decreased (Figure 8B After). These results suggest that TwX improved bowel movements and stool condition.

The most common effects perceived by participants taking TwX were “I wake up well in the morning” and “I can now sleep soundly” for both males and females. These results were obtained after combining the responses of participants who were bad but improved after taking TwX (Improved) and those who had no issues but still felt improvement after taking TwX (Originally good, but better now).

Participants reported improved QOL due to less anxiety and depression, less frustration and stress, and less daytime sleepiness (Figure 9). Improvements in appearance, such as “I now have a lighter complexion,” were also reported by nearly 40% of the females.

In addition, there were reports of subjective symptoms outside of the questionnaire items, such as “The smell of stools is no longer harsh. I no longer have smelly flatulence” and “My stools were soft in most case before taking the supplement. A few days after starting it, my stools became normal.” showing an improvement in the intestinal environment. There were also reports of reduced snoring and fatigue, improved skin quality and allergic diseases, and feeling healthy (Table 1).

Bowel Movement Frequency (BMF) and stool hardness are associated with the composition of the GM; changes in BMF and stool hardness, especially constipation, have been reported to be common in patients with dementia [39]. A large population-based cohort study of Japanese subjects found that a lower BMF and stool consistency were associated with a higher risk of dementia, respectively [40]. In addition, constipation precedes key signs of neurodegenerative diseases, such as AD, Parkinson’s disease [41,42], and Lewy body dementia [43], suggesting that improving defecation and stool condition is one way to prevent neurodegenerative diseases.

Sleep disturbances are common in AD and have a significant impact on the patients themselves and their caregivers [44]. Changes in sleep patterns are a normal symptom of aging, manifesting in the form of fragmented sleep, nocturnal arousals, and increased tendency to sleep during the day. In dementia, sleep patterns have been noted to deteriorate further [45]. Cross-sectional and longitudinal studies have reported an association between poor sleep quality and cognitive decline [46,47]. Furthermore, objective sleep fragmentation and circadian rhythm disturbances using actigraphy have been shown to increase the risk of mild cognitive impairment (MCI) and dementia in healthy older adults [48]. Improving sleep quality is an important factor in AD, as epidemiological studies have reported that up to 45% of AD patients have sleep disturbances [49].

TwX not only improved defecation and stool quality, but participants also reported improved sleep quality. The results of this study are based on participants self-reporting using a questionnaire survey; therefore, the level of accuracy cannot be verified. However, the fact that the participants took TwX and were aware of the improvement in their QOL is very significant. Although these results were obtained in healthy subjects, the results suggest that, by improving defecation and sleep quality, TwX may help to prevent AD and improve the QOL of AD patients. Further blinded clinical trials involving human subjects are necessary to obtain an accurate verification of these results.

## 3. Materials and Methods

### 3.1. Materials

TwX consists of the following active ingredients: L-glutamine (34.6 wt%), ascorbic acid (34.2 wt%), L-cystine (18.2 wt%), coenzyme Q10 (3.6 wt%), succinic acid (3.6 wt%), fumaric acid (3.6 wt%), riboflavin (1.5 wt%), and niacin amid (0.7 wt%).

For the rat OPP study, TwX was dissolved with MiliQ water (Sigma-Aldrich, Tokyo, Japan) and stored at 4 °C until use. Participants who completed the questionnaire survey were provided with TwX by TIMA Japan Corporation (Osaka, Japan).

Orthophenyl phenol was purchased from Tokyo Chemical Industry Co., Ltd. (Tokyo, Japan). Medetomidine, midazolam, and butorphanol were purchased from Nippon Zenyaku Kogyo Co., Ltd. (Fukushima, Japan), Astellas Pharma Inc. (Tokyo, Japan), Meiji Seika Pharma Co., Ltd. (Tokyo, Japan), respectively. Isoflurane was purchased from Fujifilm Wako Pure Chemicals Co., Ltd. (Osaka, Japan).

### 3.2. Animals

Wistar male rats obtained from Japan SLC (Shizuoka, Japan) were used in this study. Rats were maintained in a temperature- and humidity-regulated room (23 ± 3 °C, 50 ± 10%) on a 12 h light–dark cycle and allowed free access to food and water unless otherwise mentioned. All experimental procedures were approved by the Animal Committee of the Gifu University Graduate School of Medicine. Student’s *t*-test was used for statistical analysis unless otherwise mentioned.

### 3.3. Rat OPP Oxidative Stress Loading Model

Male Wistar rats were purchased at 14 weeks old and acclimated until 16 weeks old. Body weight and water consumption were recorded and, at 17 weeks old, the rats were divided into 4 groups to ensure they had similar average body weight: Normal, TwX, OPP, and TwX + OPP (n = 5).

The normal group received Milli-Q orally once a day from 17 weeks old and was used as a negative control. The OPP group received Milli-Q orally once a day from 17 weeks old, and OPP 1 µg/kg/day dissolved in a drinking bottle was given ad libitum from 18 weeks old. This group was set as a positive control. The TwX group received TwX 20 mg/kg orally once a day from 17 weeks old. This group was used to assess the effects of TwX without OS induced by OPP. The TwX + OPP group received TwX 20 mg/kg orally once a day from 17 weeks old, and OPP 1 µg/kg/day dissolved in a drinking bottle was given ad libitum from 18 weeks old. This group was used to assess the preventative potential of TwX on OS, induced by OPP, by pre-administering TwX.

OPP was administered for 2 weeks and MilliQ or TwX for 3 weeks in the corresponding groups. At 20 weeks old, plasma was collected from all rats and subjected to OS and antioxidant capacity measurements.

Rats underwent whole-body perfusion with saline solution using a modified method of Gage et al. [50]. First, a 50 mL syringe was connected to a fixative tubing and syringe needle and filled with ice-cold saline. The rats were administered intraperitoneally with medetomidine (0.15 mg/kg)–midazolam (2 mg/kg)–butorphanol (2.5 mg/kg) solution, and anesthesia was maintained with isoflurane inhalation as needed. Rats were placed on a dissecting table after they no longer responded to pain stimuli.

A lateral incision of 5–6 cm was made beneath the rib cage, through the integument and abdominal wall. The liver was separated from the diaphragm. The diaphragm was then incised along the entire ribcage length to expose the pleural cavity. After cutting through the thorax to the clavicle, any tissue attached to the heart was carefully trimmed. The tip of the sternum was clamped using a hemostat and placed over the head to allow for a clear view of the major vessels.

A needle attached to the tube was inserted into the left ventricle and passed through the ascending aorta. The tip was made visible through the aortic wall so that it did not reach the aortic arch, where the brachial artery and carotid artery diverge. A hemostat was used to secure the heart and prevent the needle from leaking. Finally, the right atrium was incised to provide an outlet. Then, saline solution was injected, and adequate perfusion was confirmed.

The rats were then dissected, and fecal samples were collected from the ileocecal area by feces collection kit (Techno Suruga Lab, Shizuoka, Japan).

### 3.4. Measurement of Oxidative Stress and Antioxidant Capacity in the Blood

To measure OS in the rat plasma, the Diacron-Reactive Oxygen Metabolites (d-ROMs) test was performed according to the kit’s instructions. The d-ROMs test quantifies in-vivo OS by capturing hydroperoxide (ROOH), a metabolite produced when ROS and free radicals oxidize body components.

To measure the antioxidant capacity in the blood and body of the rats, the BAP test (Diacron-Reactive Oxygen Metabolites) and the OXY adsorption test (Diacron-Reactive Oxygen Metabolites) were performed according to the kit’s instructions.

The BAP test evaluates the antioxidant capacity of a sample by measuring its ability to reduce trivalent iron Fe^3+^ ions to divalent iron Fe^2+^ ions. The OXY adsorption test measures the a sample’s ability to scavenge hypochlorous acid (HClO), one of the most potent ROS produced by white blood cells.

### 3.5. Questionnaire Design

Questionnaires and stool sample collection before and after taking TwX were conducted by Eyez, Inc (Tokyo, Japan). Fifty-seven pairs of male–female volunteers, recruited online, who had no incidence of cancer, diabetes, hypertension, dementia or allergic diseases participated in the survey.

On the website, participants were presented with the purpose of the study, the dosing instructions for TwX, and stool sampling instructions. Each person then completed a preliminary questionnaire to self-assess their lifestyle, defecation status, and physical condition.Completion of the questionnaire was regarded as consent to participate in the study.

After completing the questionnaire, participants were instructed to take TwX (13.25 mg/kg) orally at least 20 min before breakfast once a day for 4 weeks. During this period, participants did not change their lifestyle except for taking TwX, and all participants completed the 4-week intake period.

After 4 weeks of taking TwX, participants completed another questionnaire about their lifestyle, defecation, and physical condition, and fecal samples were collected using a feces collection kit (Techno Suruga Lab, Shizuoka, Japan).

Data collection was performed by Eyez, Inc. and data are available on the company’s website (https://www.eyez.jp/media/2018_6_Twendee.pdf (accessed on 24 October 2023)) and also available in detail on the TIMA website (https://www.twendee.com/files/twendee/study-results/twendee-gut-microbiome.pdf (accessed on 24 October 2023)). This study made secondary use of these data with permission granted to the authors by both Eyez, Inc. and TIMA.

### 3.6. Sample Collection and DNA Extraction

For both the rat and human fecal samples, DNA extraction and bacterial flora analysis of the collected fecal samples were contracted to Takara Bio Inc.

Genomic DNA was isolated using the NucleoSpin Microbial DNA Kit (MACHEREY-NAGEL, Düren, Germany). Approximately 500 µL of stored fecal samples were placed into a microcentrifuge tube containing 100 µL Elution Buffer BE. The mixture was then placed into the NucleoSpin Beads Tube with Proteinase K, which was subjected to mechanical beads’ beating for 12 min at 30 Hz in the TissueLyzer LT. The subsequent extraction procedure was performed per the manufacturer’s instructions. Extracted DNA samples were purified using the Agencourt AMPure XP (Beckman Coulter, Brea, CA, USA).

### 3.7. 16S rRNA Analysis

Two-step PCRs were performed on the purified DNA samples to obtain sequence libraries. The first PCR was performed to amplify using a 16S (V3–V4) Metagenomic Library Construction Kit for NGS (Takara Bio Inc., Kusatsu, Japan) with primer pairs of 341F (5′-TCGTCGGCAGCG TCAGATGTGTATAAGAGACAGCCTACGGGNGGCWGCAG-3′) and 806R (5′-GTCTCGTGGGCTCGGAGATGTGTATAAGAGACAGGGACTACHVGGGTWTCTAAT-3′) corresponding to the V3–V4 region of the 16S rRNA gene. The second PCR was carried out to add the index sequences for Illumina sequencer with barcode sequence using the Nextera XT Index kit (Illumina, San Diego, CA, USA). The prepared libraries were subjected to the sequencing of paired-end 250 bases using the MiSeq Reagent Kit v3 on the MiSeq (Illumina).

Microbiome analysis was performed using the open-source bioinformatics pipeline, Quantitative Insights Into Microbial Ecology (QIIME; Version 1.8.0). In the representative sequence preparation, CD-HIT-OTU (Version 0.0.1) was used to create an operational taxonomic unit (OTU).

The detailed workflow is as follows: first, sequenced paired-end reads were assembled to construct contigs. In the next step, chimeric contigs were removed by applying the CD-HIT-OTU algorithm and the remaining contigs were clustered into OTUs with 97% sequence similarity. To acquire taxonomic information for each OTU, representative sequences were assigned to Greengenes 16S rRNA database (g_13.8) using an RDP classifier (Version 2.2).

For the calculation of number of bacterial species detected and the diversity index, an analysis was performed using QIIME (alpha-diversity of QIIME for the diversity index, Version 1.8.0), and CD-HIT (Version 4.6) was conducted to create OTUs. Taxonomy was classified using Takara Bio original database (based on RDP).

The number of observed OTUs was calculated to assess the sensitivity of bacteria detection and the degree of bacteria diversity within each sample. A comparative analysis was conducted of alpha-diversity and the relative abundance of bacteria for each sample. To analyze alpha-diversity, a Monte Carlo simulation two-sample t-test was applied to the rat and human samples. For t-test statistic calculation, a maximum read number that corresponded to the point at which all observable bacteria could be detected for all samples was used. Subsequently, the false discovery rate (FDR) was applied to obtain the *p*-value. To compare the relative abundance of bacteria in the samples, a Monte Carlo simulation two-sample t-test was applied. Subsequently, FDR was applied to obtain the *p*-value. For human data, the *p*-value was adjusted by applying the Bonferroni method.

The data from the human samples mentioned above are available on the website of Eyez, Inc. (https://www.eyez.jp/media/2018_6_Twendee.pdf (accessed on 24 October 2023)) and TIMA establishment (https://www.twendee.com/files/twendee/study-results/twendee-gut-microbiome.pdf (accessed on 24 October 2023)) in detail. This study conducted a secondary analysis of the existing data with the permission of Eyez, Inc. and the TIMA establishment.

## 4. Conclusions

The present study demonstrated that the gut microbiota can be greatly altered by short-term exposure to OS. TwX reduced OS, regulated GM, and improved QOL. Dementia is generally believed to result from long-term exposure to OS, but it is possible that GM is significantly altered first. Therefore, TwX may initially improve the gut microbiota, and then show various other effects later. This suggests that GM may also be involved in the dementia-preventive effects that TwX has previously shown.

## Figures and Tables

**Figure 1 ijms-24-16642-f001:**
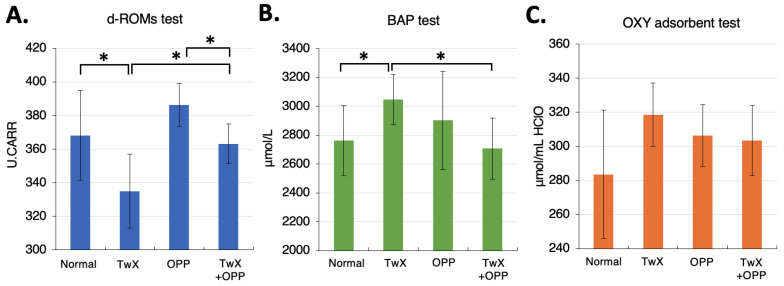
(**A**) Blood oxidative stress (d-ROMs test), (**B**) blood antioxidant potential (BAP test), and (**C**) body antioxidant potential (OXY adsorbent test) after rat OPP oxidative stress load. d-ROMs test, BAP test, OXY adsorbent test by Diacron International Srl, Grosseto, Italy. Values in the graph represent the mean ± SD. *: *p* < 0.05 (Student’s *t*-test).

**Figure 2 ijms-24-16642-f002:**
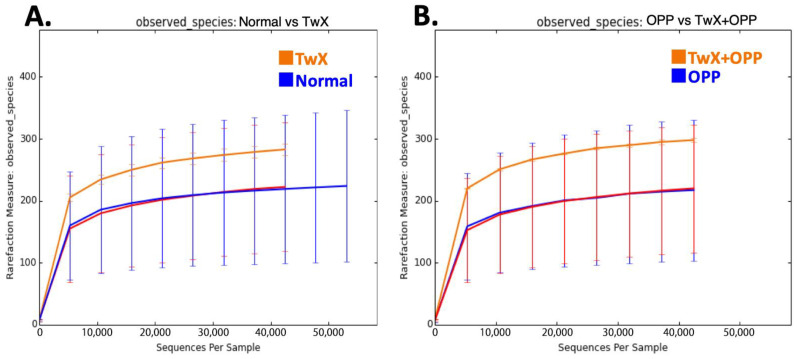
Comparison of bacterial diversity in the rat intestine. Rarefaction curves observed based on observed species (OTUs) value are shown. (**A**) Normal group vs. TwX group, (**B**) OPP group vs. TwX + OPP group.

**Figure 3 ijms-24-16642-f003:**
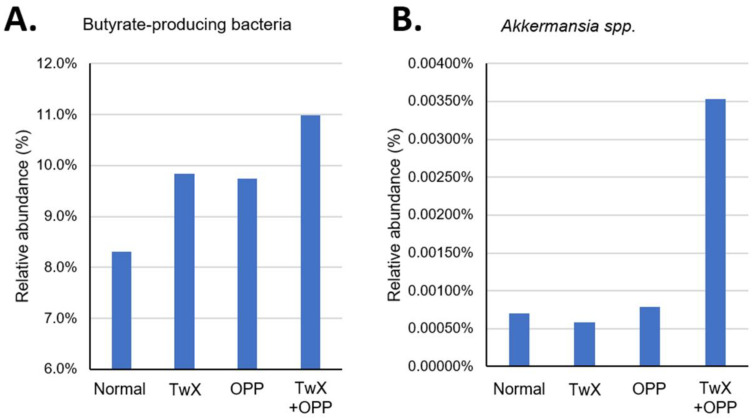
Changes in relative abundance of beneficial bacteria in the rat intestine before and after taking Twendee X (20 mg/kg/d). (**A**) Butyrate-producing bacteria. (**B**) *Akkermansia* spp. No significant difference was observed.

**Figure 4 ijms-24-16642-f004:**
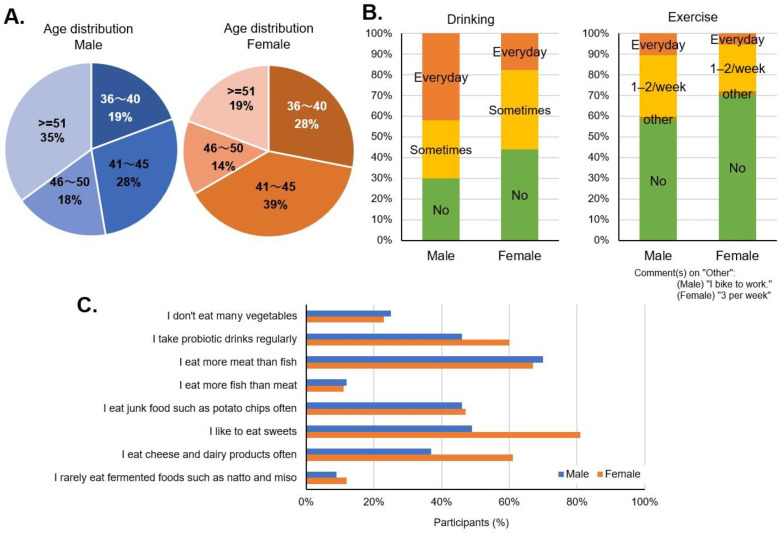
Participant survey response. Before starting to take Twendee X, participants self-reported their (**A**) age, (**B**) drinking and exercise habits, and (**C**) eating habits, by sex.

**Figure 5 ijms-24-16642-f005:**
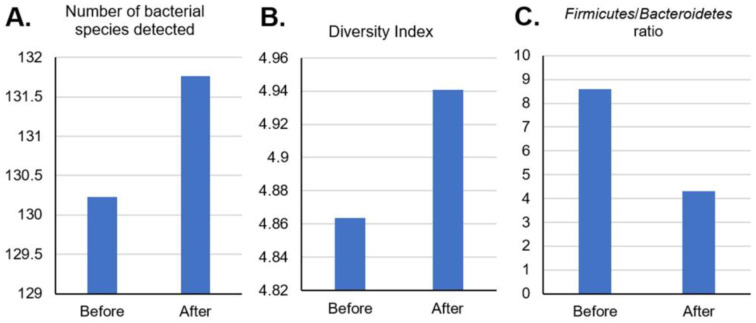
Levels of participants’ gut microflora before and after taking Twendee X. (**A**) Number of bacterial species detected, (**B**) diversity index, and (**C**) *Firmicutes*/*Bacteroidetes* ratio.

**Figure 6 ijms-24-16642-f006:**
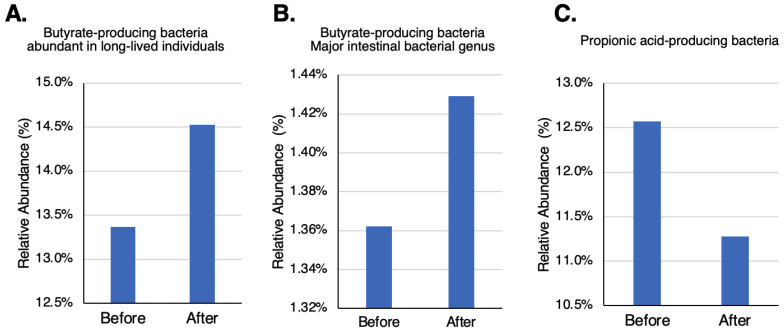
Changes in beneficial bacteria in the human intestine after taking Twendee X. (**A**) Butyrate-producing bacteria: the genera *Roseburia*, *Coprococcus*, *Lachnospira*, and *unclassified Lachnosiraceae*, abundant in long-lived individuals. (**B**) *Eubacterium* and *Clostridium*, mostly butyrate-producing genera. (**C**) *Bacteroides*, mostly propionic acid-producing genus. No significant change was observed.

**Figure 7 ijms-24-16642-f007:**
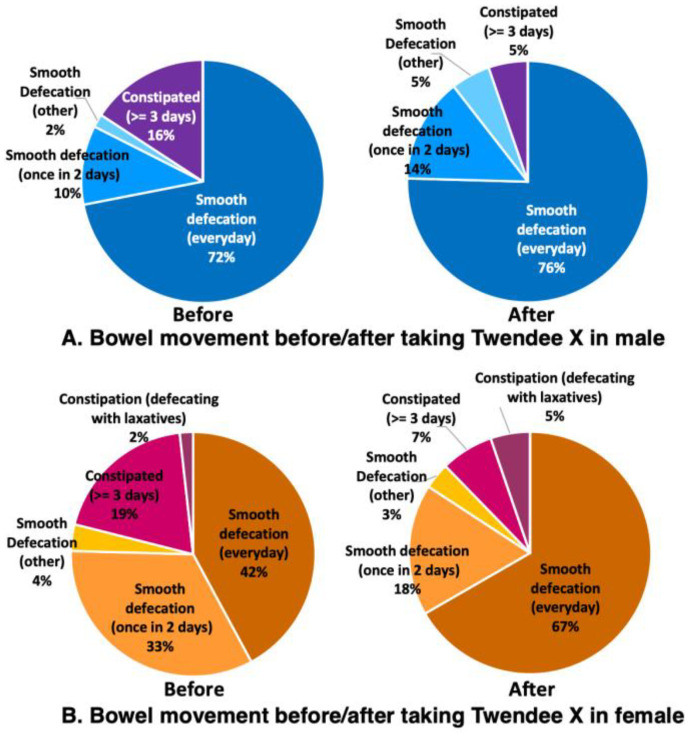
Changes in bowel movements before and after taking Twendee X for 4 weeks. (**A**) Male and (**B**) female participants.

**Figure 8 ijms-24-16642-f008:**
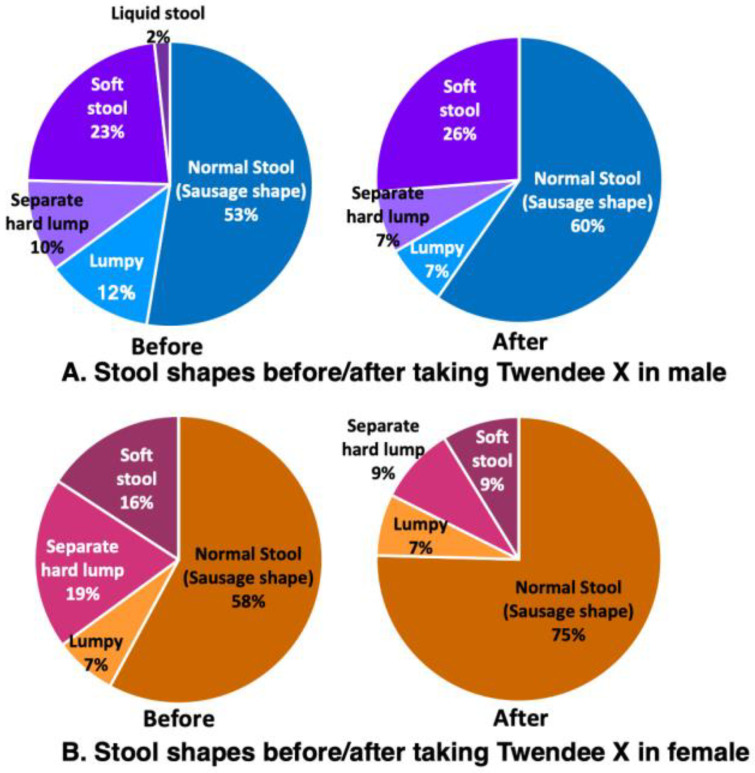
Changes in stool shape before and after taking Twendee X for 4 weeks. (**A**) Male and (**B**) female participants.

**Figure 9 ijms-24-16642-f009:**
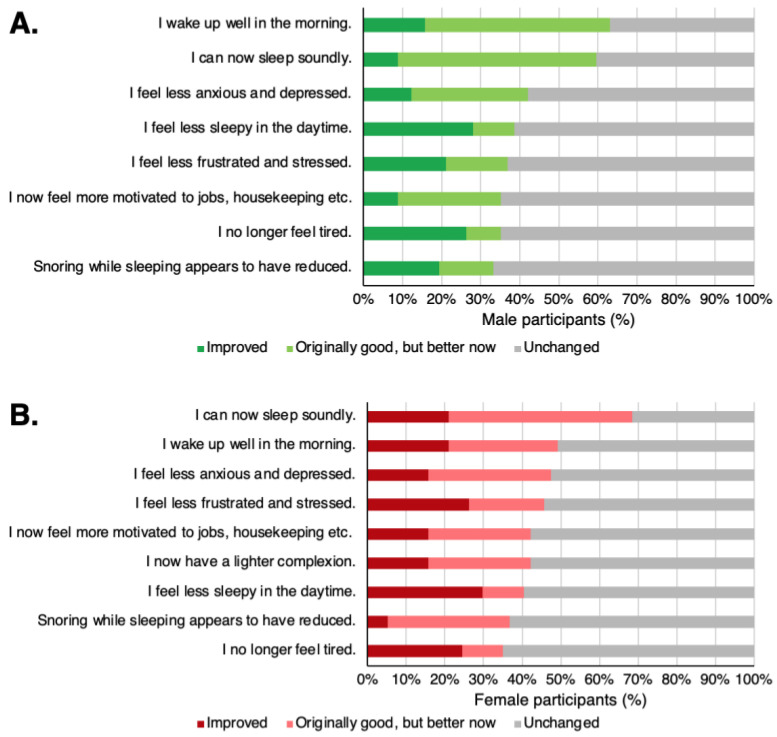
Notable perceived physical changes after taking Twendee X (TwX). (**A**) Male, (**B**) female. The chart shows the conditions that were considered below normal before but were perceived to have improved after taking TwX (Improved), those that were not noticeable before but were perceived to have improved after taking TwX (Originally good but better now), and those that was not noticeable originally and had no perceived change (Unchanged).

**Table 1 ijms-24-16642-t001:** Improvements after taking TwX, as indicated in the comments of the survey.

**Male Participants**
- Taking the supplement regularly helped me wake up early in the morning without an alarm. - Before taking the supplement, my stools were soft in most case. However, a few days after starting it, my stools became normal. - I think my physical condition is better than before. - I generally feel a little healthier. - My partner told me that I am snoring less frequently lately. - I was able to sleep better at night and wake up better in the morning, feeling less tired.
Female participants
- The smell of stools is no longer harsh. I no longer have smelly flatulence. - I now have a daily bowel movement. In addition, my hay fever has improved, and I seem to be falling asleep earlier at night. - I wake up feeling better in the morning and can get up more easily. My skin is less dry and the problems have disappeared. I believe my skin spots have lightened. - My bowel movements have improved considerably. Gum boils caused by pus in the gums, not mouth ulcers, have stopped appearing. - I have noticed an improvement in my skin and no longer experience as many breakouts. I can now stay awake until night without needing to take naps. - I feel healthy. People now frequently comment on how much better my complexion looks.

## Data Availability

All available data can be found at the URLs in the text.

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
