# Peer review of "Controlling Gut Microbiota by Twendee X® May Contribute to Dementia Prevention"

_ijms, 2023, doi:10.3390/ijms242316642_

Round 1

Reviewer 1 Report

Comments and Suggestions for Authors

The research carried out by the authors of the manuscript is devoted to an urgent scientific problem – the determination of the protective properties of the new Twendee X dietary supplement in relation to oxidative stress, the composition of the intestinal microbiota and indicators of the functional activity of the digestive and nervous systems. The information obtained by the authors is of high importance for the practical application of the original dietary supplement. The results shown by the authors of the manuscript are of unconditional interest to researchers in the field of neurobiology, gastroenterology and neurology. Therefore, the conducted research is certainly relevant for publication in the journal IJMS.

The relevance and justification of the hypothesis formulated by the authors for the conducted research are given in detail and clearly in the section "Introduction". The study was carried out at a good methodological level, and the use of research methods is described in detail. The results of the research are objectively and clearly presented in the manuscript and discussed in detail in the "Discussion" section with an emphasis on current data on the direction chosen by the authors.

However, the manuscript in its current form contains the following some disadvantages:

1. The relationship between oxidative stress and the composition of the microbiota studied in animals and the indicators of the functional activity of the digestive and nervous systems studied in healthy volunteers have an indirect relationship to dementia and Alzheimer's disease. Therefore, it is advisable to remove the indication of the presence of a preventive effect for these diseases in the drug Twendee X from the name, annotation and conclusion.

2. The authors need to change the wording in the manuscript to focus on the study of antioxidant properties, the composition of the intestinal microbiota and indicators of the functional activity of the digestive and nervous systems.

3. It is necessary to reflect the use of statistical methods in the relevant subsection of the chapter "Materials and methods".

4. Section 2.3. it is more correct to call "Effects on the functional activity of the digestive and nervous systems", because the term "Effects on quality of life" is rarely used in healthy volunteers.

5. It is better to move the first paragraph from the chapter "Conclusion" to the chapter "Results and discussion"

Author Response

We wish to thank you for the comments.

We have provided our responses below to all of your comments. Thank you in advance for your understanding.

  1. The relationship between oxidative stress and the composition of the microbiota studied in animals and the indicators of the functional activity of the digestive and nervous systems studied in healthy volunteers have an indirect relationship to dementia and Alzheimer's disease. Therefore, it is advisable to remove the indication of the presence of a preventive effect for these diseases in the drug Twendee X from the name, annotation and conclusion.

    RESPONSE: Thank you for pointing this out.
    Since the results of this study were obtained upon administration of TwX, we believe that there is no problem with the inclusion of TwX. We hope you will agree with us here.

  1. The authors need to change the wording in the manuscript to focus on the study of antioxidant properties, the composition of the intestinal microbiota and indicators of the functional activity of the digestive and nervous systems.

    RESPONSE: Thank you for pointing this out.
    These results would certainly suggest some connection to the digestive and nervous systems. However, since the results of this study may represent only a very small part of the above mechanism, we are expressing them in this manner. We hope you will agree with us here.

  2. It is necessary to reflect the use of statistical methods in the relevant subsection of the chapter "Materials and methods".

    RESPONSE: Thank you for pointing this out.
    We have added the description to Line 349.

  3. Section 2.3. it is more correct to call "Effects on the functional activity of the digestive and nervous systems", because the term "Effects on quality of life" is rarely used in healthy volunteers.

    RESPONSE: Thank you for pointing this out.
    Indeed, the results in healthy individuals were shown in this case. Defecation and sleep quality play a role in quality of life. We believe that such improvements in healthy people may well be seen in patients with MCI and AD. This is why we have chosen this title. We hope you will understand.

  4. It is better to move the first paragraph from the chapter "Conclusion" to the chapter "Results and discussion"

    RESPONSE: Thank you for pointing this out.
    We have moved the relevant paragraph to Line 323 of the Results and Discussion.

Reviewer 2 Report

Comments and Suggestions for Authors

The manuscript titled “Controlling gut microbiota by Twendee X®ï¸Ž also contributes to dementia prevention”, explores an intriguing connection between Twendee X®ï¸Ž, gut microbiota, and dementia prevention, several aspects need refinement to strengthen the scientific rigor and ensure the validity of the conclusions drawn. Here are some comments to consider improving the manuscript,

1.           The title suggests a strong causal relationship between controlling gut microbiota with Twendee X®ï¸Ž and dementia prevention. However, the abstract does not adequately justify such a conclusive statement. The findings should be presented with more cautious language, acknowledging the complexities of dementia etiology.

2.           The introduction provides a comprehensive overview of the gut microbiota, but it lacks clarity on the specific knowledge gap or research question prompting this study. A more explicit statement of the study's purpose and hypotheses would enhance the introduction.

3.           The rationale for using the OPP-induced OS model is not thoroughly justified. More information on why this specific model was chosen and how it reflects the long-term OS elevation observed in AD patients would strengthen the study's foundation.

4.           The manuscript heavily relies on previous studies to establish Twendee X®ï¸Ž's efficacy. While referencing past research is crucial, the manuscript should prioritize presenting new data and emphasize how the current study contributes to the existing body of knowledge.

5.           The manuscript asserts a link between alterations in gut microbiota and dementia, but the specific mechanisms are not adequately explained. A more in-depth exploration of the biological pathways involved would enhance the manuscript's scientific rigor.

7.           The interpretation of GM diversity results lacks depth. The authors should elaborate on the significance of alterations in bacterial diversity, considering the potential implications for overall gut health and the development of neurological disorders.

8.           The design of the human study, particularly the questionnaire survey, raises concerns about subjectivity and recall bias. Objective measures or additional validation methods should be considered to strengthen the reliability of the reported outcomes.

9.           The concluding remarks highlight the self-reported nature of the findings, emphasizing the limitations. However, the discussion could benefit from a more critical evaluation of these limitations and suggestions for future research directions.

Comments on the Quality of English Language

Minor editing of English language required

Author Response

We wish to thank you for the comments.

We have provided our responses below to all of your comments. Thank you in advance for your understanding.

  1. The title suggests a strong causal relationship between controlling gut microbiota with Twendee X®ï¸Ž and dementia prevention. However, the abstract does not adequately justify such a conclusive statement. The findings should be presented with more cautious language, acknowledging the complexities of dementia etiology.

    RESPONSE: Thank you for pointing this out.
    As you mentioned, dementia is a very complex disease, and we have changed the title of the paper to reflect this.

  2. The introduction provides a comprehensive overview of the gut microbiota, but it lacks clarity on the specific knowledge gap or research question prompting this study. A more explicit statement of the study's purpose and hypotheses would enhance the introduction.

    RESPONSE: Thank you for pointing this out.
    We have added an explanation to Line 76 to clarify the purpose and hypothesis of the study.

  3. The rationale for using the OPP-induced OS model is not thoroughly justified. More information on why this specific model was chosen and how it reflects the long-term OS elevation observed in AD patients would strengthen the study's foundation.

    RESPONSE: Thank you for pointing this out.
    OPP is approved as a preservative for citrus fruits, and products with the peel, such as jams, exist and are naturally palatable. In light of this, we wanted to use a model of elevated oxidative stress under more natural conditions in this study, so we used OPP. Additional explanation have been made to the text Line 90 regarding the above.

  4. The manuscript heavily relies on previous studies to establish Twendee X®ï¸Ž's efficacy. While referencing past research is crucial, the manuscript should prioritize presenting new data and emphasize how the current study contributes to the existing body of knowledge.

    RESPONSE: Thank you for pointing this out.
    We have added an explanation to the Conclusion regarding this matter.

  5. The manuscript asserts a link between alterations in gut microbiota and dementia, but the specific mechanisms are not adequately explained. A more in-depth exploration of the biological pathways involved would enhance the manuscript's scientific rigor.

    RESPONSE: Thank you for pointing this out.
    We agree with you. However, there are many biological pathways based on this study that are not yet clear and need to be investigated further. Your understanding would be greatly appreciated.
    I have referred this matter to Line169 at this time.

  6. The interpretation of GM diversity results lacks depth. The authors should elaborate on the significance of alterations in bacterial diversity, considering the potential implications for overall gut health and the development of neurological disorders.

    RESPONSE: Thank you for pointing this out. We have added an explanation to Line 125 regarding the importance of changes in bacterial diversity.

  7. The design of the human study, particularly the questionnaire survey, raises concerns about subjectivity and recall bias. Objective measures or additional validation methods should be considered to strengthen the reliability of the reported outcomes.

    RESPONSE: Thank you for pointing this out.
    As you mentioned, we have such concerns about this human study. Therefore, we hope to conduct a clinical trial in the future. For this paper, the above matters are clearly stated on line 323. We hope you will agree with us.
  8. The concluding remarks highlight the self-reported nature of the findings, emphasizing the limitations. However, the discussion could benefit from a more critical evaluation of these limitations and suggestions for future research directions.

    RESPONSE: Thank you for pointing this out.
    We have added the future research direction to Line328.

Round 2

Reviewer 2 Report

Comments and Suggestions for Authors

The authors have addressed my concerns, and the manuscript has been improved.